# Information Transmission in G Protein-Coupled Receptors

**DOI:** 10.3390/ijms25031621

**Published:** 2024-01-28

**Authors:** Roger D. Jones

**Affiliations:** European Centre for Living Technology, University of Venice, 30123 Venice, Italy; roger.jones@unive.it

**Keywords:** G protein-coupled receptors, information theory, precision medicine, angiotensin II receptor, adrenergic receptor, barcode

## Abstract

G protein-coupled receptors (GPCRs) are the largest class of receptors in the human genome and constitute about 30% of all drug targets. In this article, intended for a non-mathematical audience, both experimental observations and new theoretical results are compared in the context of information transmission across the cell membrane. The amount of information actually currently used or projected to be used in clinical settings is a small fraction of the information transmission capacity of the GPCR. This indicates that the number of yet undiscovered drug targets within GPCRs is much larger than what is currently known. Theoretical studies with some experimental validation indicate that localized heat deposition and dissipation are key to the identification of sites and mechanisms for drug action.

## 1. Introduction

Most papers on G protein-coupled receptors (GPCRs) start out with some variation of the sentence, “GPCRs are the largest class of receptors in the human genome and constitute about 30% of all drug targets [1]”. The foreseeable practical applications of this class of proteins are well established. The long-term practical and scientific applications may be even more interesting, however. GPCRs are molecular microprocessors, or perhaps more accurately “nanoprocessors,” that transmit, process, and compare information among and about the environments on each side of the cell membrane and within the membrane itself [2,3]. GPCRs are important examples of molecular computation that may shed light on the mechanisms of information processing in all adaptable biological processes. If that is true, then it may be scientifically and practically profitable to regard GPCRs within the context of general information theory and statistical physics, two disciplines with connections to broad sets of general principles. In this article, I outline recent experimental, simulation, and theoretical efforts to shed light on general principles of GPCR action.

Figure 1 is a schematic of the GPCR complex. The core of the complex is composed of seven transmembrane (7TM) alpha helices that form a barrel. There is an eighth intracellular helix that plays an important role in information transmission. A ligand in the extracellular region can bind to a pocket in the barrel and allosterically affect the conformation of the intracellular portion of the core. The intracellular portion of the complex is composed of a collection of molecular switches that can take two forms: (1) a $G_{\alpha }$ switch that involves the $G_{\alpha }$ subunit of a G protein and (2) a collection of phosphorylation sites can form phosphorylation dephosphorylation cycles [4]. The phosphorylation sites form a barcode [5,6] that transmits information to β arrestin (βarr), which directs downstream responses to the ligand.

Even at this basic level, the picture of the GPCR complex is mysterious and unsatisfying. Intuitively, we see that a small amount of information is associated with the binding of a ligand to the GPCR, and perhaps the concentration of the ligand is translated into a larger amount of information that requires a barcode to store and transmit all the response options. Precision/personalized/stratified medicine attempts to identify the individual prognosis and targeted treatment at the right time for the right patient, or at least for smaller and more homogeneous groups [7,8,9]. The nuances of the information transmission across the membrane must be clear.

The study of information transmission is taking place on three scales. The largest scale studies take place at the level of the human organism, the clinical level (e.g., [10,11]). Here, approved drug treatments are applied to patient populations and advanced statistical tools are used to tease out the individual patient characteristics that respond differentially to specific treatments. This is a coarse approach to precision drug discovery.

At the smallest scales are detailed observations of GPCR structure (e.g., [6,12]). Here, portions of the GPCR complex are studied in isolation. This may involve, for instance, isolation of the C tail of the GPCR [6] or isolation of the GPCR core [12]. Often, synthetic nanobodies are used to mimic missing parts of the complex. Function and interaction among the complex of components is inferred from the structural observations.

Assay experiments (e.g., [13,14]) take place on intermediate scales. Here, GPCR complexes that are intact, or at least chimeric, are treated with ligands and the downstream responses are observed. Precision drug targets in the complex must be inferred.

Proper characterization of information transmission in the control of downstream response requires that the observations on all three scales be glued together into a single coherent picture. If we take physics as an example, this glue is provided by theory that is able to fill in gaps in the picture. Recently a theory has been developed, the BOIS Model [15], that may be a candidate for filling in some of the unobserved gaps in the multi-scale observations. The model combines standard principles of statistical mechanics [16] with speculations on natural selection to make a series of predictions, some of which are testable and some of which lie in the gaps of the unobservable.

This article collects and organizes recent theoretical results and existing observations on information transmission by GPCRs for a non-mathematical audience. Although the consequences of the results may be consequential for clinical practice and drug design, these consequences are touched on only lightly.

## 2. Observations of Information Transmission across the Cell Membrane

### 2.1. Direct Measurement of Information Transmission in GPCRs

Information capacity can be measured from assay experiments. However, an important challenge to the measurement of information flow in these experiments is removing signal from noise. Accurate measurements require the application of multiple ligand concentrations to a single cell [14,17]. Ref. [14] focused on the muscarinic acetylcholine receptor (M3R). The muscarinic receptor-induced calcium response measured in individual HEK293 cells was repeatedly stimulated with the ligand acetylcholine. Using this approach, single cell assays in human embryonic kidney 293 (HEK293) cells found a capacity greater than two bits of information [14,18].

These results are in contrast with some previous studies using cell populations that provided lower values for the information capacity of the GPCR pathways [19]. Lower capacity measurements were a consequence of increased noise due to variable responses among cells and the fact that individual cells were only exposed to a single value of ligand concentration [14,17].

### 2.2. Observations of Information Transmission in Assay Experiments

Bias between two downstream response pathways was examined in Ref. [13]. The authors focused on two receptors, the adrenergic receptor β2AR and the angiotensin II receptor $AT_{1A}R$. Eleven different ligands were applied to β2AR, whereas ten were applied to $AT_{1A}R$. The measured downstream responses were β-arrestin (βarr) recruitment to the GPCR as well as cAMP for β2AR and $IP_{1}$ for $AT_{1A}R$ for the $G_{\alpha }$ pathway. To reduce noise in βarr response, the C terminus of the human β2AR was replaced with the C-terminal tail of the V2 vasopressin receptor tail.

In addition to the bias observations this study identified various values of half maximal effective concentration $\left(EC_{50}\right)$ for the responses. The $EC_{50}$ for each response can be used to identify the ligand concentration at which the response is activated. This is in agreement with Ref. [14] where information about the ligand concentration was identified in the information transmission. More recent assay studies have been completed but they have not yet been studied within the context of this article [12].

### 2.3. Observations of Allosteric Mechanisms of Information Transmission

The two signaling pathways, $G_{\alpha }$ and βarr mediate distinct physiological effects [12,20]. In the prototypical angiotensin II receptor, for instance, $G_{\alpha }$ coupling increases blood pressure, whereas βarr coupling promotes heart protection [21,22,23]. In opioid receptors, the $G_{\alpha }$ pathway confers pain relief whereas the βarr path may be associated with side effects such as tolerance, dependence, addiction, constipation, and respiratory depression [24,25,26,27,28].

The GPCR protein is composed of seven alpha helices arranged in a barrel conformation and one intracellular helix (Figure 2A). The helices themselves are somewhat rigid whereas the connections between the helices are flexible. The determination of which parts of the protein are flexible and which are rigid is determined by the ratio of the local bond strength to the background energy fluctuations [29,30,31,32,33,34]. At thermal equilibrium at room temperature, the background energy fluctuations are at about 0.5 kcal/mol. The amount of energy available in a single ATP is about 12 kcal/mol ([35], [Sec. 15.2]). Typical covalent bond energies are around 80 kcal/mol and greater, whereas ionic, hydrogen, and hydrophobic interaction energies are typically around 5 kcal/mol ([36], Chapter 8). Van der Waals interactions are typically 0.5–1 kcal/mol [35]. Parts of the protein with bond energy less than the fluctuation energy are unstable leading to flexibility. Parts of the protein where individual bonds, or collections of bonds, that have bonding energies greater than the fluctuation energy are rigid. The flexible and rigid parts of a protein are not fixed. If the background fluctuation level changes energy, then flexible parts of the protein can become rigid or rigid parts of the protein can become flexible. Recent observations have discovered a major hydrogen-bond network that bathes the ligand and transmits information allosterically [12].

Another part of the GPCR, the C tail, is very flexible at room temperature and can take on many different conformations ([37,38,39]). The flexible/rigid nature of the complex conformations indicates that the complex can occupy multiple active quasi-stable states (Figure 2) [40].

Typically, binding of $G_{\alpha }$ and βarr to the intracellular pocket of the GPCR is mediated by helices 6, 7, 8, and 5 [41] and the intracellular loops [42]. The 7TM conformation is altered from its baseline configuration (Figure 2A) when a ligand and a G protein form a ternary bond (Figure 2B) with the transmembrane portion of the GPCR [20]. This altered conformation transmits information from the extracellular environment to the intracellular environment. The G protein responds by activating downstream responses. The typical allosteric signature of this event is the rotation and extension of helix 6 [43]. Since only downstream responses triggered by the G protein are activated, this event is known as $G_{\alpha }$ bias after the relevant α subunit of the G protein. It has recently been found from mutation studies that receptor sites $N111^{3.35}A$ and $N294^{7.45}A$ induce biased signaling to $G_{\alpha }$ and βarr, respectively, [12]. AngII is a balanced agonist that activates both $G_{\alpha }$ and β-arrestin signaling pathways. AngII can be modified to generate AT1R-biased agonists, which could preferentially activate either signaling pathway [21,44,45].

The binding of the beta arrestin (βarr) to the GPCR is observed to have more conformations than G-protein binding [6] as well as having more interactions with the membrane environment [46]. The βarr interacts simultaneously with the core GPCR in a ternary reaction [47,48,49]. Two distinct binding processes for βarr have been identified, dubbed “core-engaged” (Figure 2(C2)) and “tail-engaged” (Figure 2(C1)) for recruitment to the core and tail, respectively, [40,48]. This is illustrated in Figure 2C,D where a G protein occupies the core pocket in Figure 2D that is occupied by βarr in Figure 2(C2). The conformation of the βarr is different in the two binding paths [50]. Interestingly, βarr may continue to trigger downstream responses even after dissociation from the GPCR [39,51,52]. A typical signature for the βarr core-binding (Figure 2C) is the rotation and extension of helices 7 and 8 [43].

### 2.4. Information Transmitted to the Barcode

The barcode structure indicates that much more information may be stored in the barcode than the approximately two bits of information transmitted to the barcode [3,5,6,12,38,53,54,55,56,57]. As suggested in Ref. [14], this extra information may be about the details of the ligand concentration. This information may not be observed in the experiments [12,13,14] because of the focus on only the $G_{\alpha }$ pathway and a single βarr recruitment pathway. In many cases, in these experiments, chimeric receptors were used that completely replaced the C tail of the receptor with a foreign C tail [13]. Except for the recruitment site, the phosphorylation sites on the foreign tail need not correspond with sites on the native tail. The use of the foreign tail removes information-storage capacity in the tail. The number of downstream pathways observable from a chimeric receptor would be two.

## 3. Theoretical Model for the Behavior of Information in Biological Switches

Biological processes require a flow of energy and matter through the processes in order to maintain themselves. More specifically, they require a flow of entropy, or equivalently, free energy, to keep the processes running in nonequilibrium steady states (NESSs) far from thermal and mechanical equilibrium. In most systems, entropy flow is generated directly by solar photon flow or by consumption of food that is derivative and downstream of the solar flow. The flow typically manifests itself as adenosine triphosphate (ATP) and adenosine diphosphate (ADP), or other nucleotide, concentrations driven far from their equilibrium values. This nonequilibrium condition can then drive a multitude of metabolic processes into NESSs. Natural selection then selects metabolic processes best able to survive in the environment generated by physical conditions and the milieu of other processes that are occurring simultaneously.

Natural selection also selects for adaptability, the ability to change the processes in response to shifting conditions external to the process. If many processes are adaptable, then a global NESS may never be achieved, or, at least, take a very long time to achieve. Adaptability requires management of information flow. Conditions external to the process must be measured, processed, and acted upon. In biological systems, as in engineered systems, information processing and transmission is performed by switches [4,58]. These commonly take the form of phosphorylation/dephosphorylation sites or cycles. Other types of switches, such as GTPase switches, are also possible [4]. Both are present in GPCRs.

Recently, an ab initio theoretical model of biological information and entropy processing was developed that was able to make specific verifiable predictions [15,59]. This model, dubbed Bag of Independent Switches (BOIS), is illustrated schematically in Figure 3. In the model, a set of biological switches embedded in a flexible protein background is seated in a heat bath of temperature *T*. The switches are somewhat rigid, whereas the background is composed of flexible parts of the GPCR receptor. This picture is similar to the rigidity/flexibility of the transmembrane GPCR core we discussed previously. The model does not depend on the geometrical details of the GPCR complex. This allows the model to be addressed with traditional tools of statistical mechanics [16].

The foundational assumptions of the model are:Natural selection selects for processes that maximize information storage and transmission [58].The rate of entropy production is maximized [60].

With these assumptions and the simple schematic in Figure 3, the model predicts [15,59]:1.Ligand-bound receptors can be found in one of three switch states: inactive, active/off, and active/on:(a)An inactive state in which there is no flux of the ligand-bound receptor moving between binary switch configurations. Since there is no chemical flux, there is also no heat deposition in an inactive switch. A switch in equilibrium with the background heat bath is inactive.(b)An active/off state in which the ligand-bound receptor can be found in one of the two switch configurations. There is chemical flux of the ligand bound receptor between the two switch configurations, a dephosphorylated state for example. Heat is deposited in this switch that is dissipated to the heat bath. This switch is far from equilibrium.(c)An active/on state in which the ligand-bound receptor can be found in the other of the two switch configurations. There is chemical flux of the ligand-bound receptor between the two switch configurations; for example, a phosphorylated state. Heat is deposited in this switch that is dissipated to the heat bath. This switch is far from equilibrium.2.The chemical fluxes in all switches are equal. The heat dissipation in each switch can be variable.3.If the receptor has never been in contact with the ligand and the ligand concentration is zero, then all switches are in the inactive state. As the ligand concentration increases, the switches activate one at a time. The total number of active switches is a measure of the ligand concentration. As the ligand concentration decreases from large values to small values, the ligand/receptor dissociation constants are smaller due to the stabilization of the GPCR complex by active switches.4.Information on the concentration of the ligand is contained in the number of active switches.

These predictions are specific enough to be tested with an experiment. The last prediction, for instance, is in agreement with information transmission experiments that indicate that more than one bit of information is transmitted across the membrane and that this information is associated with the ligand concentration [14]. Assay experiments also provide good tests for the model.

## 4. Assay Observations

First, consider early assay experiments [13]. The earlier experiments generated dose-response and bias curves (Figure 4) for multiple ligands bonding with the adrenergic receptor and with angiotensin II receptor (Appendix A Table A1). Bias was measured between two pathways, $G_{\alpha }$ pathway and a βarr recruitment pathway for each receptor. The summary results are displayed in Table 1.

As can be seen in Figure 4C, the first switch to activate as the ligand concentration increases is the $G_{\alpha }$ for the adrenergic receptor (black circle at (1,0)). The ligands, for this ligand concentration, are $G_{\alpha }$ biased. As the ligand concentration increases further, some ligands activate the βarr recruitment pathway. The response is balanced for this value of ligand concentration (black circle at (1,1)).

The picture is a bit more complicated for the angiotensin II receptor. Some ligands activate the $G_{\alpha }$ switch first (black circle at (1,0)). Then, all those ligands activate the βarr recruitment switch (black circle at (1,1)). Other ligands activate the βarr switch at the ligand concentration that corresponds to activation of the second switch (black circle at (0,1)). These ligands are βarr biased. For this set of ligands, the first switch is not observed to be activated. The BOIS Model can provide a possible interpretation for this behavior.

The information flow through the GPCRs in these experiments is predicted to 2.58 bits by the BOIS Model. This is in agreement with experimental observations in similar assays [14].

The BOIS Model predicts that switches can exist in three states: an inactive state that does not dissipate heat, an active/off state that dissipates heat but may not trigger a downstream response, and an active/on state that can trigger a downstream response. All switches are inactive for receptors that have never been exposed to the ligand. The switches activate one at a time as the ligand concentration increases. This can be seen in the assay data of Figure 4A,B and Table 1 where the order at which the switches activate as well as the $EC_{50}$ of the activation is displayed as a log10 of the molar ligand concentration. We assume that switches in state active/off do not trigger downstream responses. The separation of the switches to active/off and active/on is displayed in Figure 5 and in Figure 6 and Figure 7. More recent studies [12,61] have observed ligands with $G_{\alpha }$ bias for the angiotensin II receptor. This is displayed in Figure 7F,G.

It can be seen in the bias curves of Figure 4C,D that active/on switches are activated to a common receptor concentration as predicted by the BOIS Model. All receptor concentrations are normalized to the same concentration for a specific receptor.

For these assays in which the plethora of barcode states is not present, we see that the $EC_{50}$s only identify two ligand concentrations. This is in agreement with two-response observations in Refs. [13,62] and with the BOIS Model for two responses [59].

## 5. Discussion

We can use the concept of information capacity [58] to measure our understanding of GPCR computation. Capacity is the maximum amount of information that can be transmitted by a system. The capacity of the GPCR in Figure 1, for example, is very large. The barcode is capable of storing many bits of information. Much of this information comes from measurements of the ligand concentration (Section 3 Prediction 3). The source of the information on whether switches are in the active/off state or the active/on state is not yet known.

The level at which this information is currently used clinically is very small in comparison with the capacity. Take, for example, the angiotensin II receptor. It is known that the ligand angiotensin II triggers the angiotensin II receptor to increase blood pressure. This is one bit of information. The treatment for high blood pressure is to block the receptor either with angiotensin-converting enzyme inhibitors or angiotensin receptor blockers. Standard renin-angiotensin-system inhibitor blood-pressure treatment manipulates one bit of information out of the entire capacity of the angiotensin II receptor.

The next higher use for angiotensin II receptor information is biased signaling [12,61]. It is known that the $G_{\alpha }$ pathway is responsible for increasing blood pressure, the βarr pathway has positive effects on the heart. A goal is to block the $G_{\alpha }$ pathway and not the βarr. Progress has been made in understanding the environment within the core that interacts with the ligand to bias the signaling either toward the βarr pathway or the $G_{\alpha }$ pathway. When this is understood better, it will lead to significant clinical benefit with significant savings of life. Yet, this usage of information is still a tiny fraction of the GPCR capacity.

An important finding from this tranche of research is a consequence of three switch states. Both inactive and active/off states do not seem to be observable. Only the active/on state seems to be able to generate downstream response, at least in the case of simple bias. This means, for example, that the inactive state Figure 7A and the active/off state Figure 7B for the angiotensin II receptor are indistinguishable by observation of downstream response, but may behave quite differently from a drug dose. The BOIS Model allows some visibility into when a downstream response is not observed because the switch is either inactive or active/off.

The next higher use of information is at the system level. It is observed that the angiotensin II receptors in the kidney play a role in both systemic blood-pressure reduction, but also in local control of kidney performance [63]. This process is barely understood at all. It is unclear how or how much the information capacity is used to manage this process. Yet, this is one one more important medical challenge.

It is apparent that if natural selection designed the angiotensin II receptor system to only manage blood pressure, kidney flow, and heart health, then natural selection greatly over designed the receptor. It is using a small amount of the capacity observed in Refs. [5,6]. This is an important practical mystery.

An important insight emerged from recent experiments and their theoretical interpretation. Heat dissipation that generates fluctuation gradients on fast time scales seems to be an important component of NESSs [64,65,66]. Heat dissipation is important in the BOIS model for activating switches. Heat deposition and dissipation seem to play an important role in controlling downstream response.

An observational consequence of the importance of heat dissipation is that temporal fluctuations play key roles in the details of the barcode programming. Static and cold structural observations may miss important functional insights. This is probably well known, but this line of research may supply hints on what is missing and where to look for it.

One question we have not addressed here is the role of partial agonism in the BOIS model. Observations indicate that partial and full agonists generate different receptor conformations [67]. Early indications from more detailed representations of the BOIS model indicate that partial agonists can be associated with active/off states. This is a topic of active investigation.

Another unsolved question is the mystery of missing information. It has been shown that, within the context of the BOIS Model, the ligand concentration is encoded in the number of active switches in the intracellular region. It is not yet known what information from the extracellular space encodes the active on and off states. The active switch states certainly affect downstream response [6]. The mechanisms for deciding whether an active state is on or off are being investigated theoretically within the context of the BOIS Model.

Another issue is the kinetics of ligand association and dissociation. It is well known that ligands bind to receptors in a ternary reaction with a G protein [20]. The picture is that the ligand first binds to the receptor with a somewhat weak bond and then the G protein binds with the receptor stabilizing the weak ligand bond making the ligand binding much tighter than the original weak bond. Detailed balance does not hold in NESSs, so the dissociation will possibly take a different path than association. This means that the ligand/receptor dissociation constant $K_{D}$ is different when the ligand concentration is increasing rather than decreasing. In fact, the dissociation from the fully bound ternary state can be much slower than the association. Something similar may be happening with phosphorylation/dephosphorylation sites.

There are observations that receptors can be activated in the absence of a ligand [68]. Nothing in the BOIS theory so far explicitly provides an explanation for this.

The next level of mysteries that emerge from experiments and theory takse us beyond the practical horizon of current medical capabilities. They take us into the theoretical foundations of both biology and physics. The Second Law of Thermodynamics, one of the foundational laws of physics [16], states that the energy and matter in the universe will spread out until there is nothing left but chaos. How does life appear in such a dismal universe? Life can emerge from fluctuations in the flow of entropy [69]. Information and entropy are very closely related [58]. Recent experiments seem to indicate that the interaction of information and the Second Law play an important role in the evolution of adaptable life.

## Figures and Tables

**Figure 1 ijms-25-01621-f001:**
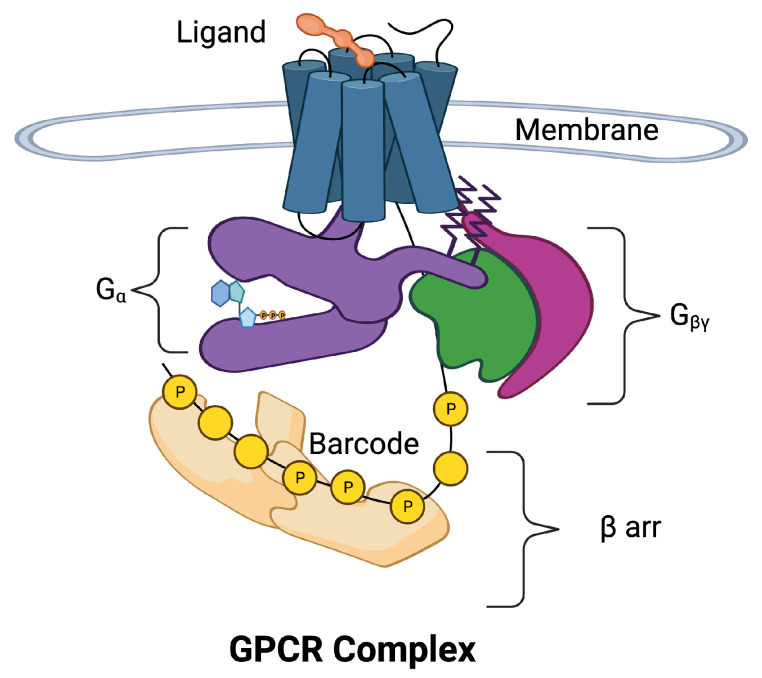
The 7-transmembrane GPCR is illustrated with blue cylinders representing the seven α helices that span the cell membrane. The extracellular ligand (orange) binds to the binding site of the GPCR inducing movement in the α helices. The helices allosterically alter the conformation of the intracellular domains of the GPCR complex. The intracellular portion of the complex has been separated for visibility. Two pathways may be activated, the $G_{\alpha }$ pathway (purple) and the βarr pathway (tan). The $G_{\alpha }$ subunit is a part of the G protein also composed of subunits β (green) and γ (magenta). The βarr pathway is composed of additional response pathways determined by phosphorylation sites on the C tail of the GPCR and intracellular loops that form a barcode that encodes signals for downstream processes.

**Figure 2 ijms-25-01621-f002:**
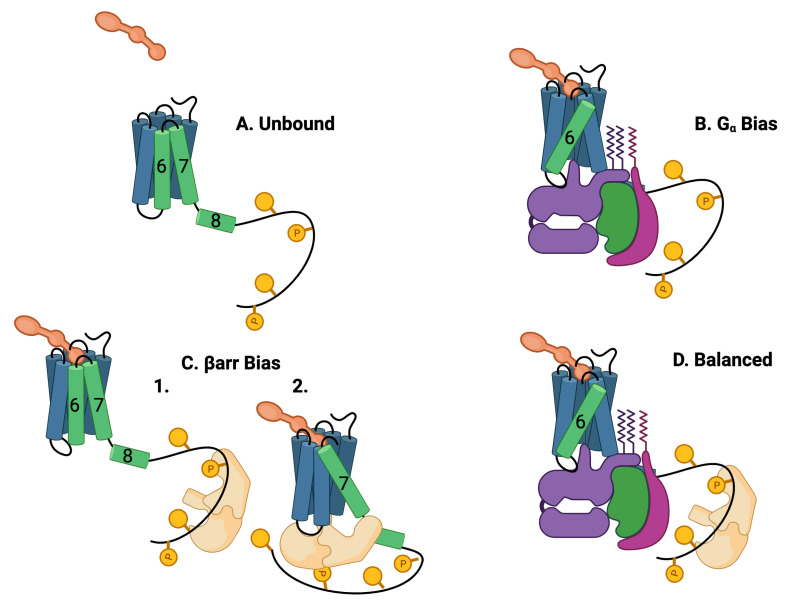
Five theoretically possible GPCR complex configurations, assuming that $G_{\alpha }$ and βarr cannot occupy the intracellular core pocket simultaneously. (**A**) **Unbound:** The GPCR is composed of seven transmembrane (7TM) alpha helices and one intracellular helix labeled (8). The alpha helices are rigid and are connected by flexible amino-acid chains. The C tail of the GPCR may be quite long and contain multiple phosphorylation sites. Particularly important helices are in green. (**B**) $G_{\alpha }$
**bias:** The ligand (orange) and the G protein (purple, green, and violet) may bind to the 7TM in a ternary reaction in which both the ligand and the G protein alter the 7 TM conformation. The most common conformational indicator is the rotation and extension of helix 6 (green). This configuration is G biased. (**C**) β**arr bias:**
βarr may bind to the C tail of the GPCR (1) or βarr may also replace the G protein in the intracellular binding pocket (2). The conformation is characterized by a rotation and extension of helix 7. This conformation blocks the G-protein downstream response pathways. The C tail may also bind to the βarr that may select βarr downstream responses. This conformation is βarr biased. (**D**) **Balanced:** Another possibility is that the G protein and the βarr are both bound to the 7TM, the G protein bound to the 7TM pocket, and the βarr bound to the C-tail phosphorylation sites.

**Figure 3 ijms-25-01621-f003:**
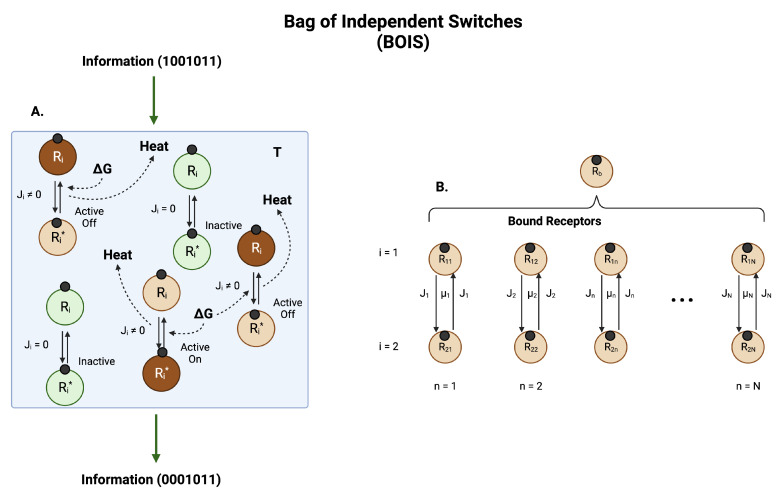
**Information and Entropy Flow.** (**A**) A collection of switches in a heat bath at temperature *T*. Each switch is localized to a position on a receptor. The switches can be distinguished by their location. For definiteness, we imagine the switches are phosphorylation sites on a ligand-bound protein receptor, although other types of switches, such as GTPase switches, are common. Ligands are represented by small black circles. A receptor with site *i* dephosphorylated is designated $R_{i}$. If the site is phosphorylated, the receptor designation is $R_{i}^{*}$. If no free energy *G* is input to a switch, the switch equilibrates with the heat bath. Switches in thermal equilibrium (green circles) experience detailed balance, which means there is no chemical flux $J_{i}$ between the states $R_{i}$ and $R_{i}^{*}$. More specifically, it means that the net flux in each of the arrows in a switch in equilibrium is zero. There is also no net heat dissipated to the heat bath. A switch in this state is labeled *inactive*. If free energy *G* created by the nonequilibrium imbalance of adenosine triphosphate (ATP) and adenosine diphosphate (ADP) is input to a switch, then a net flux $J_{i}\geq 0$ is generated in the switch and heat is deposited in the heat bath at a rate $J_{i}\mu_{i}$ where $\mu_{i}$ is the amount of heat generated by a switch when the receptor moves from dephosphorylated to phosphorylated and back. Switches in which there is finite flux are designated as *active*. A switch in which the receptor spends most of its time in the phosphorylated state $R_{i}^{*}$ of an active switch is an active-on switch. If the receptor spends most of its time in the dephosphorylated state $R_{i}$ of an active switch then the switch is labeled as active-off. The set of switches can support the transmission and manipulation of information through the system. Active-on receptors are brown in the figure. (**B**) The total probability of finding a bound receptor is distributed among the *N* possible switches. The switches can be arranged in a one-dimensional array and ordered ascendingly according to the amount of heat $\mu_{i}$ dissipated in one transit of the receptor through the switch. Each switch exists in one of the states of (**A**).

**Figure 4 ijms-25-01621-f004:**
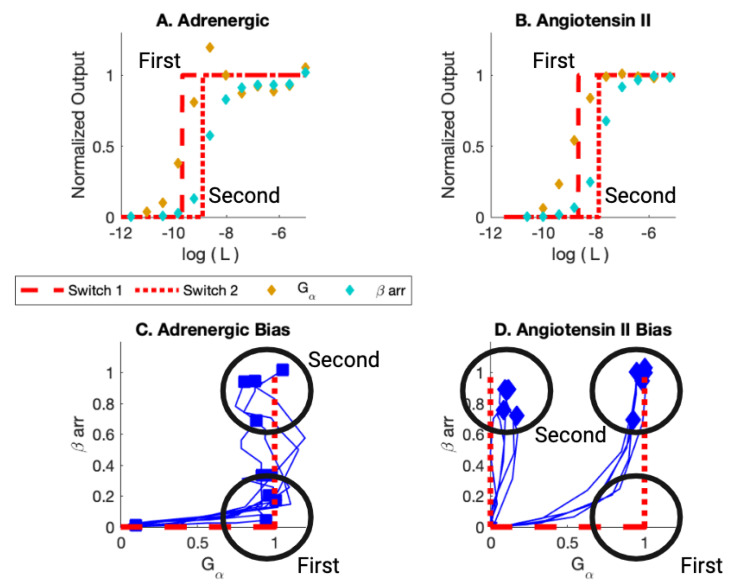
(**A**) Adrenergic Receptor with Formoterol as Ligand. Here, *L* is the ligand concentration. The theory is displayed in red. The dashed line is the simulation of the activation of the first switch. The dotted curve is the activation of the second switch. The yellow markers are the observed assay dose response for the $G_{\alpha }$ pathway. The cyan markers are the observed assay dose response for β arr. (**B**) Angiotensin II Receptor with Angiotensin II as Ligand. (**C**) Bias Plot for All Ligands for Adrenergic Receptor. The simulation results are displayed in red. Note that some ligands are $G_{\alpha }$ biased; their endpoints lie close to the $G_{\alpha }$ axis. Other ligands are balanced; their endpoints lie at (1,1). For the $G_{\alpha }$ bias, the first switch activated is the $G_{\alpha }$ switch in the on state. The second switch is activated in the βarr off state. For Balanced, the first switch is activated in the $G_{\alpha }$ on state and the second is activated in the βarr on state. (**D**) Bias Plot for All Ligands for angiotensin II Receptor. This plot illustrates balanced bias and β arr bias. For balanced bias, the first switch activates $G_{\alpha }$ in the on state and the second switch activates βarr recruitment in the on state. For β arr bias, the $G_{\alpha }$ is activated in the off state when the first switch is activated and then the βarr is activated in the on state when the second switch is activated. Figure reproduced from [59].

**Figure 5 ijms-25-01621-f005:**
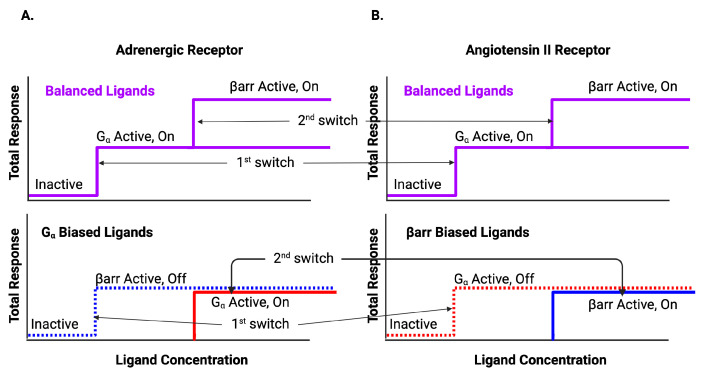
BOIS interpretation of biased data. Ligands can be classified by whether the response is balanced or biased. The upper curve is the response for balanced ligands. The lower curve is the response for biased ligands. The system supports two switches that activate sequentially. For low initial values of ligand concentration, all switches are inactive. As ligand concentration increases, the first switch is activated. For balanced ligands, the first switch observed to be activated is $G_{\alpha }$, whereas for biased ligands, the first switch can be either $G_{\alpha }$ (**A**) or βarr (**B**). For biased ligands, the first switch is activated in the off state. For the adrenergic receptor (**A**), the $G_{\alpha }$ switch does not need to be activated in order to activate the βarr switch into the off state for biased ligands. After the first switch activates, but before the second switch activates, half the switches are active and half are inactive. As ligand concentration increases further, the second switch activates activating the βarr recruitment pathway for balanced ligands and the remaining inactive pathways for the biased ligands. Note that the BOIS model predicts that if the ligand concentration is lowered from its maximum, the activated switches do not deactivate. Therefore, the number of active switches is a measure of the maximum ligand concentration. Figure reproduced from [59].

**Figure 6 ijms-25-01621-f006:**
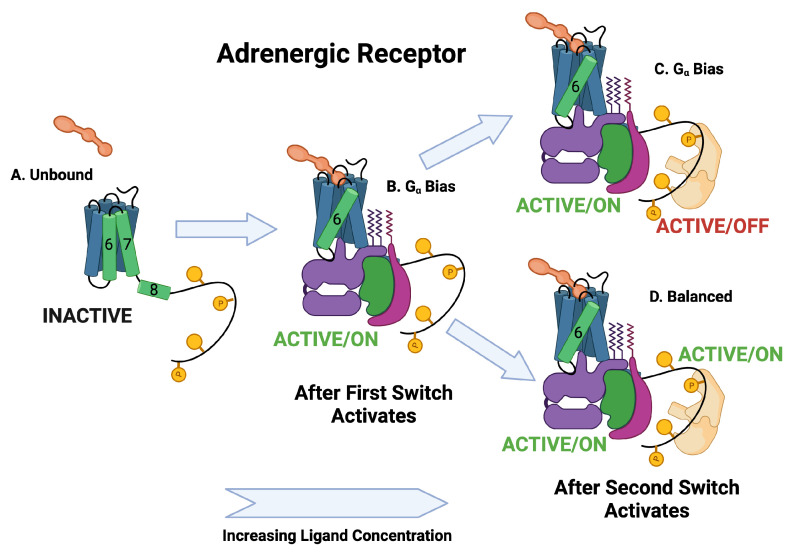
Schematic of adrenergic receptor activation. The $G_{\alpha }$ switch is the first switch to activate and it activates into the active/on state. The second switch to activate is the βarr recruitment switch. Some of the ligands activate the active/off βarr switch and some activate the active/on βarr switch.

**Figure 7 ijms-25-01621-f007:**
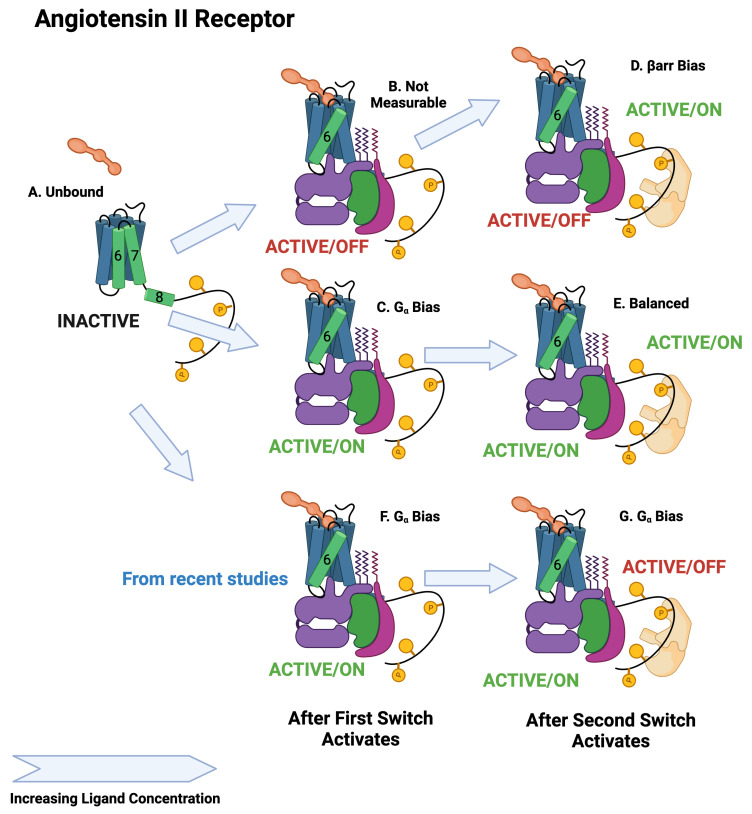
Schematic of angiotensin II activation. The first switch to activate is the $G_{\alpha }$ switch. Depending on the ligand, the activated state will be active/off or active/on. As in the case of the adrenergic receptor, the second switch to activate is the βarr recruitment switch, which activates in the active/on state. The upper path is βarr biased, whereas the middle path is balanced. The lower path is $G_{\alpha }$ biased [12].

**Table 1 ijms-25-01621-t001:** **Summary of Assay Data.** Two receptors were tested with several ligands. The detailed results and the ligands are given in Appendix A and Ref. [13]. The purple row is for those outcomes in which both the $G_{\alpha }$ and the βarr recruitment switches are activated to the on state by the ligands. The red cells indicate assays in which the $G_{\alpha }$ switch is activated to the on state by the ligands, but the βarr recruitment switch is activated to off. The blue cells indicate assays in which the $G_{\alpha }$ switch is activated to off by the ligands, but the βarr recruitment switch is activated to on. The yellow cell indicates a ligand that did not turn on either the $G_{\alpha }$ or the βarr switch. The $G_{\alpha }$ column indicates the mean logarithm of the molar ligand concentration at which the $G_{\alpha }$ switch turns on. The βarr column indicates the concentration at which the βarr switch turns on. An *X* indicates that the switch did not turn on. The Order columns indicate the order in which the switch turns on as ligand concentration increases. We see, for balanced ligands, that the second switch turns on at a ligand concentration approximately one order of magnitude higher than the concentration at which the first switch turns on. For the biased ligands, the order of the switch turning on is determined by comparison with the concentrations of the balanced ligands. For example, the biased $G_{\alpha }$ ligands are determined to be the second switch turning on by noting that the concentration −7.76 for the biased turn on is approximately equal to −7.95, the concentration of the second switch to turn on for the balanced ligands. Table reproduced from [59].

Adrenergic	Angiotensin II
	**Order**	$G_{\alpha }$	**Order**	β **arr**		**Order**	$G_{\alpha }$	**Order**	β **arr**
**Bal**	1st	**−9.44**	2nd	**−7.95**	**Bal**	1st	**−8.06**	2nd	**−7.14**
**Bias**	2nd	**−7.76**		X	**Bias**		X	2nd	**−6.86**
None		X		X

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
