# Peer review of "Information Transmission in G Protein-Coupled Receptors"

_ijms, 2024, doi:10.3390/ijms25031621_

Round 1

Reviewer 1 Report

Comments and Suggestions for Authors

This is an interesting article, which proposes new ways to think about, and integrate, the huge amounts of biological data we have relating to GPCRs. It certainly seems that this paper could help to set the scene for a whole field of work. It is well written and explains difficult concepts well.

My main questions are possibly out of scope for this work as the article does seem to stand on its own. Regardless I think that these following points need some acknowledgement in the text, even if they aren't incorporated into the model at this point. 

1. How does the model differentiate between partial and full agonists? This is surely another layer of information encoded by the GPCR that could have therapeutic relevance e.g. Masureel et al 2019 (Structural insights into binding specificity, efficacy, and bias of a B2AR partial agonist) shows that there are different conformational states in receptors that are stabilised by full and partial agonists.

2. Possibly building on the above point, how does the model differentiate between biological activation profiles that alter the same downstream effectors in different ways? For instance human and salmon calcitonin both promote cAMP accumulation through calcitonin receptor with similar pEC50's, but it has been shown that they differently influence the hetero-trimeric G protein (Furness et al 2016, Ligand-Dependent Modulation of G Protein Conformation Alters Drug Efficacy). Could some of the additional layer of information be from the kinetics of receptor activation (how quickly it adopts Active/ON, how long it stays like that, and then how long it is OFF before returning to ON), and then from the kinetics of the downstream effector molecules? Likewise, does the core-engaged vs tail-engaged differentiation get incorporated?

3. How well does the model take into account basally active receptors and inverse agonists and the current understanding that these receptors are basally sampling conformations that can lead to receptor activation in the absence of ligand? E.g. Wingler and Lefkowitz 2020 (Conformational basis of G protein-coupled receptor signaling versatility).

Minor points:

Page 1, Line 11: The introduction is listed as section 0, depending on the style of the journal it might be more appropriate to list this as section 1.

Figure 1, Legend line 2: "Allosteric binding site". I understand that this refers to the fact that the binding of the ligand allosterically modifies the rest of the receptor, however an allosteric binding site has a number of different connotations within pharmacology that I don't think are being referred to here. Consider changing to orthosteric binding site, or even just ligand binding site.

Page 3, Line 78: Needs a full stop after (M3R).

Page 3, Line 100: Needs a full stop after [12].

Page 4, Line 136: Needs to indicate which protein experiences these mutations are - I presume these are in the receptor, but it is possible that the mutations are in the Ga and arrestin proteins.

Page 4, Line 137: You haven't introduced the different Ga protein subtypes, so probably better to remove Gq and replace with Ga.

Page 4, Line 144: It might be easier to refer to your figure right after mentioning core-engaged and tail-engaged, for example core-engaged (Figure 2C2) and tail-engaged (Figure 2C1).

Figure 2, Legend last line: Uncapitalise t in "Tail"

Page 5, Line 157: You mention that the receptors used are often chimeric without mentioning whether this is a problem or consideration. It would be worth even a throwaway line explaining whether you consider this to be a factor in interpretation.

Page 6, Line 176: You have already used the acronym PdPCs (Page 1, line 34), but you only use that phrase twice, so you could just remove the acronym. Likewise GTPC (line 177) is only referred to once, so could be removed. This also applies to Page 14, line273 and 24 with ACE, ARB and RASi.

Page 10, Line 256: Two full-stops after BOIS model.

Figure 7: "From recent studies" seems out of place. Probably uneccessary?

Page 13, Line 271: "receptor to increase in blood pressure" - does not need the "in" between increase and blood.

Page 14, Line 290: You mention that this is a gap that could be filled by theory, but did not attempt to describe the theory. Possibly mention whether this is work for future.

Overall very enjoyable article, many thanks for writing this.

Author Response

RDJ: Dear Reviewer 1: Thank you for your insightful and detailed review.

  1. How does the model differentiate between partial and full agonists? This is surely another layer of information encoded by the GPCR that could have therapeutic relevance e.g. Masureel et al 2019 (Structural insights into binding specificity, efficacy, and bias of a B2AR partial agonist) shows that there are different conformational states in receptors that are stabilised by full and partial agonists.

RDJ: The following paragraph has been added to the discussion section:

One question we have not addressed here is the role of partial agonism in the BOIS
model. Observations indicate that partial and full agonists generate different receptor
conformations [ 67 ]. Early indications from more detailed representations of the BOIS
model indicate that partial agonists can be associated with active/off states. This is a topic
of active investigation.

  1. Possibly building on the above point, how does the model differentiate between biological activation profiles that alter the same downstream effectors in different ways? For instance human and salmon calcitonin both promote cAMP accumulation through calcitonin receptor with similar pEC50's, but it has been shown that they differently influence the hetero-trimeric G protein (Furness et al 2016, Ligand-Dependent Modulation of G Protein Conformation Alters Drug Efficacy). Could some of the additional layer of information be from the kinetics of receptor activation (how quickly it adopts Active/ON, how long it stays like that, and then how long it is OFF before returning to ON), and then from the kinetics of the downstream effector molecules? Likewise, does the core-engaged vs tail-engaged differentiation get incorporated?

RDJ: The following two paragraphs has been added to the discussion section:

Another unsolved question is the mystery of missing information. It has been shown
that, within the context of the BOIS Model, the ligand concentration is encoded in the
number of active switches in the intracellular region. It is not yet known what information
from the extracellular space encodes the active on and off states. The active switch states
certainly affect downstream response [6]. The mechanisms for deciding whether an active
state is on or off is being investigated theoretically within the context of the BOIS Model.

Another issue is the kinetics of ligand association and dissociation. It is well known
that ligands bind to receptors in a ternary reaction with a G protein [ 20]. The picture is
that the ligand first binds to the receptor with a somewhat weak bond and then the G
protein binds with the receptor stabilizing the weak ligand bond making the ligand binding
much tighter than the original weak bond. Detailed balance does not hold in NESSs, so
the dissociation will possibly take a different path than association. This means that the
ligand/receptor dissociation constant KD is different when the ligand concentration is
increasing rather than decreasing. In fact, the dissociation from the fully bound ternary
state can be much slower than the association. Something similar may be happening with 329
phosphorylation/dephosphorylation sites.

  1. How well does the model take into account basally active receptors and inverse agonists and the current understanding that these receptors are basally sampling conformations that can lead to receptor activation in the absence of ligand? E.g. Wingler and Lefkowitz 2020 (Conformational basis of G protein-coupled receptor signaling versatility).

RDJ: The following sentence has been added to the discussion section:

There are observations that receptors can be activated in the absence of a ligand [ 68 ].
Nothing in the BOIS theory so far explicitly provides an explanation for this.

Minor points:

Page 1, Line 11: The introduction is listed as section 0, depending on the style of the journal it might be more appropriate to list this as section 1.

Rdj: I am using the Latex template provided by the journal. The journal selected the heading format.

Figure 1, Legend line 2: "Allosteric binding site". I understand that this refers to the fact that the binding of the ligand allosterically modifies the rest of the receptor, however an allosteric binding site has a number of different connotations within pharmacology that I don't think are being referred to here. Consider changing to orthosteric binding site, or even just ligand binding site.

Rdj: I removed the word “allosteric.”

Page 3, Line 78: Needs a full stop after (M3R).

RDJ: Done. Thank you.

Page 3, Line 100: Needs a full stop after [12].

RDJ: Done

Page 4, Line 136: Needs to indicate which protein experiences these mutations are - I presume these are in the receptor, but it is possible that the mutations are in the Ga and arrestin proteins.

RDJ: I added the word “receptor” to indicate that the mutations were done on the receptor.

Page 4, Line 137: You haven't introduced the different Ga protein subtypes, so probably better to remove Gq and replace with Ga.

RDJ: Done

Page 4, Line 144: It might be easier to refer to your figure right after mentioning core-engaged and tail-engaged, for example core-engaged (Figure 2C2) and tail-engaged (Figure 2C1).

RDJ: Done

Figure 2, Legend last line: Uncapitalise t in "Tail"

RDJ: Done and added a hyphen.

Page 5, Line 157: You mention that the receptors used are often chimeric without mentioning whether this is a problem or consideration. It would be worth even a throwaway line explaining whether you consider this to be a factor in interpretation.

I added the following sentences to the end of Sec. 1.

Except for the recruitment site, the phosphorylation sites on the foreign tail need not correspond with sites on the native tail. The use of the foreign tail removes information-storage capacity in the tail. The number of downstream pathways observable from a chimeric receptor would be two.

Page 6, Line 176: You have already used the acronym PdPCs (Page 1, line 34), but you only use that phrase twice, so you could just remove the acronym. Likewise GTPC (line 177) is only referred to once, so could be removed. This also applies to Page 14, line273 and 24 with ACE, ARB and RASi.

RDJ: Fixed. Thank you.

Page 10, Line 256: Two full-stops after BOIS model.

RDJ: Fixed. Thank you.

Figure 7: "From recent studies" seems out of place. Probably uneccessary?

RDJ: I removed “from recent studies” in the caption and added a reference. I left the phrase in the figure itself to indicate that the conclusion from that part of the figure was from a different study.

Page 13, Line 271: "receptor to increase in blood pressure" - does not need the "in" between increase and blood.

RDJ: Fixed.

Page 14, Line 290: You mention that this is a gap that could be filled by theory, but did not attempt to describe the theory. Possibly mention whether this is work for future.

I replaced the gap sentence with the following sentence:

The BOIS Model allows some visibility into when a downstream response is not observed because the switch is either inactive or active/off.

Reviewer 2 Report

Comments and Suggestions for Authors

Overall, the manuscript is well-written, easy to read, and contains interesting information. 

According to the claim stated in the introduction, the review attempts to describe the process of information transmission in GPCRs at three different scales – the human organism as the largest scale, GPCR molecular structure as the smallest scale, and intermediate scale, which the author defines as intact GPCRs complexes treated with ligands, and connections between these scales. Specifically, line 66 states  “This short review attempts to organize the observations on the three different scales and show how the observations might be glued together …”

Actually, only the GPCR structure scale is described comprehensively in Part 1, which includes 10 paragraphs and dozens of references. The human organism or clinical part is only slightly mentioned in the introduction and conclusion, no review in this part is provided. If the review of available information is out of scope of this review, the goal should be reformulated to avoid reader’s confusion.

My main concern is the description of the processes that describe the information flow in the intact GPCRs complexes treated with ligands, which the author named as the intermediate level. Although this part of the review is the largest, it goes around only one theory that was proposed recently by the author and it is based only on three studies and one conference proceedings paper. The conference proceedings paper and one of the three papers belong to the author of this article and both were published very recently (2023). For one experimental paper it was only mentioned that it is consistent with the author’s theory and one experimental paper was treated very comprehensively from the point of view of the author’s theory.

For this reason, this part of the manuscript cannot be considered as a review and should be either extensively complemented by description of models and studies of other authors, which the author actually mentioned in his previous paper (Jones, R.D.; Jones, A.M. Frontiers in Endocrinology 2023), or published as an opinion or an erratum to the previous paper.

Although the author's attempt to provide a new general theory can become highly impactful, it would benefit if the author presented his contributions in a more traditional form, not mixing review and justification of his own theory.

I think in the present form it cannot be considered as a review article or research article and probably the author should rethink the structure.

Minors

- Typical energy of van der Waals interactions for proteins is 0.5 - 1 kcal/mol (Berg JM, Tymoczko JL, Stryer L. Biochemistry. 5th edition. New York: W H Freeman; 2002). 

The text contains several typos, e.g. 

- “in the in the” (line 30); 

- add a dot (line 218); 

- “Fig. D” (change to Fig. 2D, line 145); 

- “Fig. C2” (change to Fig. 2C, line 146); 

- “For the adrenergic receptor A., The” (remove dot, change capital T to t, caption to Figure 5);

- To dots (line 256)

Author Response

RDJ: Dear Reviewer 2: Thank you for your insightful and detailed review.

Overall, the manuscript is well-written, easy to read, and contains interesting information. 

According to the claim stated in the introduction, the review attempts to describe the process of information transmission in GPCRs at three different scales – the human organism as the largest scale, GPCR molecular structure as the smallest scale, and intermediate scale, which the author defines as intact GPCRs complexes treated with ligands, and connections between these scales. Specifically, line 66 states  “This short review attempts to organize the observations on the three different scales and show how the observations might be glued together …”

Actually, only the GPCR structure scale is described comprehensively in Part 1, which includes 10 paragraphs and dozens of references. The human organism or clinical part is only slightly mentioned in the introduction and conclusion, no review in this part is provided. If the review of available information is out of scope of this review, the goal should be reformulated to avoid reader’s confusion.

RDJ: I removed the paragraph that claimed that the article reviews clinical results. This is the removed paragraph.

This short review attempts to organize the observations on the three different scales and show how the observations might be glued together with theory to form a more coherent and possibly useful theory. Relevant observations from the assay scale and the molecular scale are first identified. It is then shown that some of the missing observables can be predicted by theory. Finally, in the discussion, an attempt is made to apply the emergent picture at the clinical level.

RDJ: The last sentence in Sec. 1 has been replaced with:

This article collects and organizes recent theoretical results and existing observations on information transmission by GPCRs for a non-mathematical audience. While the consequences of the results may be consequential for clinical practice and drug design, these consequences are touched on only lightly.

I modified the sentence referring to review in the abstract to”

In this article, intended for a non-mathematical audience, both experimental observations and new theoretical results are compared in the context of information-transmission across the cell membrane

My main concern is the description of the processes that describe the information flow in the intact GPCRs complexes treated with ligands, which the author named as the intermediate level. Although this part of the review is the largest, it goes around only one theory that was proposed recently by the author and it is based only on three studies and one conference proceedings paper. The conference proceedings paper and one of the three papers belong to the author of this article and both were published very recently (2023). For one experimental paper it was only mentioned that it is consistent with the author’s theory and one experimental paper was treated very comprehensively from the point of view of the author’s theory.

RDJ: I replaced the word “consistent” with the words “in agreement with” in three locations in the paper that compare theory with experiment.

For this reason, this part of the manuscript cannot be considered as a review and should be either extensively complemented by description of models and studies of other authors, which the author actually mentioned in his previous paper (Jones, R.D.; Jones, A.M. Frontiers in Endocrinology 2023), or published as an opinion or an erratum to the previous paper.

Although the author's attempt to provide a new general theory can become highly impactful, it would benefit if the author presented his contributions in a more traditional form, not mixing review and justification of his own theory.

I think in the present form it cannot be considered as a review article or research article and probably the author should rethink the structure.

RDJ: I replaced the word “review” everywhere with the word “article.” I characterized the article as one that “collects and organizes recent theoretical results and existing observations on information transmission by GPCRs for a non-mathematical audience.” I will consult the editors for the proper classification of the article.

I removed the sentence:

I attempt to tie the general principles to the more immediate application of drug discovery and treatment.

from Sec. 0.

Minors

- Typical energy of van der Waals interactions for proteins is 0.5 - 1 kcal/mol (Berg JM, Tymoczko JL, Stryer L. Biochemistry. 5th edition. New York: W H Freeman; 2002). 

RDJ: I replaced that part of the text with the following:

Typical covalent bond energies are around 80 kcal/mol and greater, while ionic, hydrogen, and hydrophobic interaction energies are typically around 5 kcal/mol [36 , Chapter 8]. Van der Waals interactions are typically 0.5 − 1 kcal/mol [35 ].

Thank you.

The text contains several typos, e.g. 

- “in the in the” (line 30); 

- add a dot (line 218); 

- “Fig. D” (change to Fig. 2D, line 145); 

- “Fig. C2” (change to Fig. 2C, line 146); 

- “For the adrenergic receptor A., The” (remove dot, change capital T to t, caption to Figure 5);

- To dots (line 256)

RDJ: Fixed. Thank you.

Round 2

Reviewer 1 Report

Comments and Suggestions for Authors

Many thanks to the author for an interesting paper, and for addressing my concerns.

Reviewer 2 Report

Comments and Suggestions for Authors

The author has carefully addressed each point raised, providing modifications, clarifications, and revisions to strengthen the manuscript. I have no other questions.